# *Agaricus macrochlamys*, a New Species from the (Sub)tropical Cloud Forests of North America and the Caribbean, and *Agaricus fiardii*, a New Synonym of *Agaricus subrufescens*

**DOI:** 10.3390/jof8070664

**Published:** 2022-06-24

**Authors:** Rosario Medel-Ortiz, Roberto Garibay-Orijel, Andrés Argüelles-Moyao, Gerardo Mata, Richard W. Kerrigan, Alan E. Bessette, József Geml, Claudio Angelini, Luis A. Parra, Jie Chen

**Affiliations:** 1Centro de Investigación en Micología Aplicada, Universidad Veracruzana, Xalapa 91070, Mexico; romedel@uv.mx; 2Instituto de Biología, Universidad Nacional Autónoma de México, Mexico City 04510, Mexico; rgaribay@ib.unam.mx (R.G.-O.); evoandres@gmail.com (A.A.-M.); 3Instituto de Ecología, A. C. Carretera Antigua a Coatepec 351, El Haya, Xalapa 91073, Mexico; gerardo.mata@inecol.mx; 4Independent Researcher, Kittanning, PA 16201, USA; rwkres@windstream.net; 5Independent Researcher, Burlington, NC 27215, USA; rwkres@winfstream.net; 6ELKH-EKKE Lendület Environmental Microbiome Research Group, Eszterházy Károly Catholic University, Leányka u. 6. G. épület, H-3300 Eger, Hungary; jozsef.geml@gmail.com; 7Jardín Botánico Nacional Dr. Rafael Ma. Moscoso, Apartado 21-9, Santo Domingo 10602, Dominican Republic; claudio_angelini@libero.it; 8Independent Researcher, Via Cappuccini 78/8, 33170 Pordenone, Italy; 9Independent Researcher, Avda. Miranda do Douro 7, 5° G, 09400 Aranda de Duero, Spain; agaricus@telefonica.net; 10Facultad de Ciencias Biológicas y Agropecuarias, Peñuela, Universidad Veracruzana, Amatlán de los Reyes 94945, Mexico

**Keywords:** Agaricaceae, *Agaricus* subg. *flavoagaricus*, almond mushrooms, Caribbean, neotropics, phylogeny, taxonomy

## Abstract

*Agaricus* is a genus of fungi in the family Agaricaceae, with several highly priced edible and medicinal species. Here we describe *Agaricus macrochlamys*, a new species, in *A*. sect. *Arvenses*, sympatric and morphologically cryptic with the edible and medicinally cultivated mushroom, *A. subrufescens*. Phylogenetic analyses showed that *A. macrochlamys* is closely related to *A. subrufescens*, and that *A. fiardii* is a new synonym of *A. subrufescens*. Despite being morphologically cryptic, *A. macrochlamys* can be distinguished from *A. subrufescens* by several ITS and *tef1α* species-specific markers and a 4-bp insertion in the *tef1α* sequence. Furthermore, *A. subrufescens* is a cosmopolitan species, while *A. macrochlamys* distribution is so far restricted to Mexico, the Dominican Republic, and the United States.

## 1. Introduction

*Agaricus* L. (Agaricaceae, Basidiomycota) includes saprotrophic mushrooms of medicinal, nutraceutical and nutritional interest. The genus has a worldwide distribution comprising around 580 species [1,2,3,4,5,6]. According to Zhao et al. [7], the diversity of the genus *Agaricus* is clearly underestimated. Since 2000, taxonomic studies of the genus *Agaricus* have been very scarce in Africa, South America and Oceania, where only 17 new species have been described.

Despite its ecological and economic importance, this genus has scarcely been studied in Mexico and the Dominican Republic, unlike in the United States. Mata et al. [8] reported 32 species for Mexico; Medel et al. [9] concluded that in Mexico tropical cloud forest is the ecosystem with the greatest diversity of species of *Agaricus*; Palestina-Villa et al. [10] updated the nomenclature of the species known from Mexico. Regarding the Dominican Republic, two works with the description of 19 taxa have been recently published [6,11].

*Agaricus subrufescens* Peck is a cosmopolitan and very variable taxon described from material collected in the United States at the end of the 19th century [12]. The presence of *A. subrufescens* in Mexico and the Dominican Republic has only recently been confirmed with molecular methods by Velázquez-Narváez et al. [13] and Parra et al. [11], respectively.

The nomenclature, morphological molecular identity and intraspecific variability of wild specimens of *A. subrufescens* were discussed by Kerrigan [2,14], Parra [15], Thongklang et al. [16] and Chen et al. [17], who concluded that *A. albopersistens* Zuccher., *A. bambusae* Heinem., *A. blazei* Murrill *sensu* Heinemann, *A. brasiliensis* Wasser, M. Didukh, Amazonas and Stamets (an illegitimate name), *A. rufotegulis* Nauta and *A. rufotegulis* var. *hadriaticus* Lancon. and Nauta are all synonyms of *A. subrufescens*, the current correct name. Kerrigan [14] also reported that the samples from Hawaii were genetically the most divergent, and that the numerous heteromorphisms shared by the ITS rDNA sequences of European and American samples could reflect events of hybridization between ancient divergent populations, while Chen et al. [17] noted that the ITS sequences can be classified into three different types, based on nine informative polymorphic positions.

*Agaricus subrufescens* is a member of *A*. sect. *Arvenses* (Konrad and Maubl.) Konrad and Maubl., the only section in *A*. subg. *Flavoagaricus* Wasser. This section is characterized by medium-to-large sized basidiomata, a superous two-layered annulus with conspicuous scales or a radially arranged cogwheel on the lower surface, the context generally turning yellow when bruised or cut with an odor of aniseed or almonds, KOH and Schäffer’s reactions immediate and strongly positive, cheilocystidia generally catenulate and the hyphae of the lower surface of the annulus with inflated elements [5,15]

In our previous study, an ITS sequence of a specimen (F-2285), collected in Martinique and identified as *A. fiardii* by J-P. Fiard, the collector of the type, was obtained. This sequence was identified as the ITS type A of *A. subrufescens* [11], which implied that the type specimen of *A. fiardii* may, in fact, belong to *A. subrufescens,* particularly because the type specimen had also been collected in Martinique [18]. Coincidentally, the new species here proposed is cryptic and sympatric with *A. subrufescens,* because both of the species were collected in the same areas of the Dominican Republic and Mexico. Thus, to get an unequivocal identification of *A. fiardii*, the type specimen was re-examined; however, numerous attempts to sequence the type specimen in different laboratories using the Sanger method failed. Recently, one of the authors (J. Geml) was able to obtain a partial ITS sequence (ITS2 region), using an Illumina, and the sequence matched with *A. subrufescens*.

With the synonymy between *A. fiardii* and *A. subrufescens* now unequivocally established, we propose *A. macrochlamys* as a new species which can be distinguished from *A. subrufescens* by having five and six ITS and *tef1α* species-specific markers, respectively, and a 4-bp insertion in the *tef1α* sequence. Furthermore, *A. subrufescens* is a cosmopolitan species distributed in both the temperate and (sub)tropical regions, while the known range of *A. macrochlamys* is so far restricted to the (sub)tropical cloud forests in Mexico, the Dominican Republic and the United States.

## 2. Materials and Methods

### 2.1. Sampling and Morphological Description

The fresh specimens were collected at various localities in the province of Puerto Plata, (Dominican Republic), the state of Veracruz (Mexico) and the state of Georgia (USA). The specimens were collected from the wild, except for three of the specimens collected from cultivated strains, which are identified by the acronym CUL-IEXXXX.

The strains CA603, CA1110, IE903, IE904, IE965, IE973 and IE4012 were isolated from the wild collections Mata 753, Medel 1961, Velázquez-Narváez 1456, Mata 855, Medel and Lorea-Hernández 2249, Velázquez-Narváez 2357 and Montiel 20, respectively, and grown on PDA (potato-dextrose-agar medium). These strains are kept as a living culture in refrigeration at 4 °C at the Fungal Culture Collection biorepository of the Instituto de Ecología, Xalapa, Veracruz, Mexico.

All of the studied materials were deposited at the mycological herbaria of FACBA (Facultad de Ciencias Biológicas y Agropecuarias, Universidad Veracruzana, Córdoba, Veracruz, Mexico), JBSD (Dr. Rafael M. Moscoso, Jardín Botánico de Santo Domingo, Dominican Republic), XAL (Instituto de Ecología, Xalapa, Veracruz, Mexico) and the private herbarium of Richard W. Kerrigan. The specimens deposited in JBSD have duplicates in the private herbarium of Luis A. Parra (LAPAM).

The morphological descriptions were completed with fresh wild specimens, except for the Mexican collections Velázquez-Narváez 1456, Mata 855 and Montiel 20, in which the basidiomata, obtained from cultivation and identified as CUL-IE903, CUL-IE904 and CUL-IE4012, respectively, coming from the previously isolated strains IE903, IE904 and IE4012, were used. The dried material from CUL-IE903, CUL-IE904 and CUL-IE4012 were preserved in herbaria (see material examined). The macroscopic description includes a characterization of the pileus, lamellae, stipe, annulus and context, and the microscopic description includes a characterization of the basidiospores, basidia, cheilocystidia, pileipellis and lower surface of the annulus. The spore measurements followed the statistical method proposed by Heinemann and Rammeloo [19], measuring at least 30 spores of each collection. The temporary slides were mounted using H_2_O, KOH (5%), phloxine or Congo Red for microscopic study. The macrochemical reactions, such as Schäffer’s reaction (aniline solution plus 5% nitric acid solution) and KOH (5%) were tested on the pileus, context and stipe, as recommended by Parra [20]. The data on habit, habitat and distribution were also provided. The arrangement of the morphological characters follows Parra [20]. The herbarium acronyms are from Holmgren and Holmgren [21].

### 2.2. DNA Extraction, PCR, and Sequencing

The samples for DNA extraction were obtained from the hymenium in dried material of the wild collections or from the mycelia of living cultures for some of the Mexican collections. Different methods were used for the DNA extraction. For the Mexican samples, commercial DNA isolation kits were used: REDExtract-N-Amp™ Plant PCR Kit (Sigma-Aldrich, St. Louis, MO, USA) and the Norgen plant/fungi DNA isolation kit (Norgen Biotek Corp., Thorold, ON, Canada). For the USA sample, the DNA was extracted using the CTAB method [22]. For the Dominican Republic samples, the DNA was extracted from dry specimens employing a modified protocol, based on Murray and Thompson [23]. For the holotype of *A. fiardii*, the NucleoSpin^®^ Plant DNA isolation kit (Macherey-Nagel Gmbh & Co., Düren, Germany) was used. The DNA sequences were obtained from two regions: primers ITS1F [24], ITS4 and ITS5 [25] were used for the ITS region; and primers EF1-983F and EF1-1567R [26] for the translation elongation factor 1-alpha (*tef-1α*) region. The primers fITS7 [27] and ITS4 were used for amplifying the ITS2 rDNA region of the holotype of *A. fiardii*. The procedures for the amplification of the ITS and *tef-1α* regions followed those of White et al. [25] with some modifications [28], Chen et al. [3], Mullis and Faloona [29] and Izzo et al. [30]. The PCR products were sequenced at the Laboratorio de Secuenciación de la Biodiversidad y la Salud, Instituto de Biología, UNAM, Mexico, Eurofins (Louisville, Kentucky, USA), StabVida Inc., Portugal or Macrogen Inc., Korea. The consensus sequences were assembled by using Geneious Pro R7 or Lasergene software v. 7.1 (DNAStar, Madison, WI, USA).

### 2.3. Sampling for Phylogenetic Analyses

Two datasets were prepared for the phylogenetic analyses: the combined dataset and the ITS dataset. The combined dataset included 105 ITS and 49 *tef1* sequences, representing 44 validly named species and numerous undescribed taxa in *A*. sect. *Arvenses* and the outgroup *A. edmondoi* L.A. Parra, Cappelli and Callac. Among these, 24 ITS and 11 *tef1* sequences were newly generated from this study, and the remaining were retrieved from GenBank and were used in previous studies [2,5,31]. The sample’s origin and GenBank accession numbers are listed in Table 1. The ITS dataset included 66 sequences, of which 24 sequences were newly generated from this study: 49 sequences of *A. subrufescens* from different geographical origins and bearing different ITS types, and 17 sequences of *A. macrochlamys* were used.

### 2.4. Phylogenetic Analyses

The sequences were aligned by using MAFFT [32] for each region independently, then manually adjusted in BioEdit v. 7.0.4 [33]. Two datasets were prepared for different analyses. 

The combined dataset was used for maximum likelihood and Bayesian analyses. The maximum likelihood (ML) analysis was performed in RAxMLHPC2 v. 8.2.4 [34], as implemented on the Cipres portal [35], under a GTRGAMMA model with one thousand rapid bootstrap (BS) replicates. The Bayesian inference (BI) analysis was performed with MrBayes v. 3.1.2 [36], under a partitioned model. The combined dataset was partitioned into ITS, *tef1* intron and *tef1* coding sites. The best substitution model for each partition was inferred with the program MrModeltest 2.2 [37]: HKY + I + G for ITS; GTR for *tef1* intron sites; and SYM + I + G for *tef1* coding sites. Two runs of six Markov chains were run for one million generations and sampled every 100th generations. Burn-in was determined by checking the likelihood trace plots in Tracer v. 1.6 [38] and subsequently discarded. The outputs were displayed in FigTree v 1.4.0 (http://tree.bio.ed.ac.uk/software/figtree/ (accessed on 8 April 2022)). 

The polymorphisms in the ITS region were found to be useful to characterize the species and varieties within *Agaricus* [31,39,40]; therefore, we used them to characterize the new collections made in this study. An unusual intraspecific variability is known in *A. subrufescens*. The different ITS types (A, B or C) were identified using the polymorphisms at nine ITS positions, as described by Chen et al. [17]. Briefly, types A and B, which differ at six polymorphic positions from each other (positions 39, 122, 130, 145, 146 and 200 in Table 2) are known in the samples from the Americas and Europe. Type C, which differs at three positions from A or B (positions 269, 466, and 475 in Table 2) are known from Hawaii and Southeastern Asia. For the heteromorphic sequences at these nine positions, haploid sequences were obtained by three different methods: (i) sequencing of homokaryotic spores for WC837 [16], which is of type AB; (ii) by PCR cloning for CA487, which is of type ABC [17]; or (iii), for CA864, by interpreting the two sequences which are superimposed with an offset in the electropherograms due to a length polymorphism [17]. The latter method was applied in the present study to the Mexican strain CA603, and revealed that it was an AB type, as in the French sample CA864 (Table 2). The GenBank accession numbers that included the sequences of 57 heterokaryotic samples and nine haploid sequences are indicated in Table 2.

The ITS dataset was used for the unweighted pair group method with arithmetic mean (UPGMA) analysis. A UPGMA tree was constructed using MEGA11 [41] with default setting.

### 2.5. Species-Specific ITS Markers

Comparisons were made between the 104 sequences (ITS1 + 5.8S + ITS2) of *A*. sect. *Arvenses* used for the phylogenetic analyses. The position of a unique nucleotide (nt) in the ITS sequence of a species is indicated as follows: “xxxxxXxxxxx @ position”, where the capital letter represents the exclusive or informative character, and xxxxx represents the flanking characters. Square brackets were used for insertion or deletion (indel). It must be noted that the comparisons are made with all species of the section for which sequences are currently available; however, with more taxa, reassessment might be needed.

## 3. Results

### 3.1. Phylogenetic Analyses

The final alignment of the combined dataset included sequences of 105 specimens and 1192 characters. The phylogenetic trees generated by the ML and BI methods were very similar in topologies. The maximum likelihood (ML) tree is shown in Figure 1. *Agaricus* sect. *Arvenses* is monophyletic and four major clades are recovered, as in Cao et al. [5], and except for clade I, which is poorly supported, the remaining clades are moderately (clade II) to well-supported (clade III and clade IV). The 24 new collections are clustered in clade IV: seven of them joined the subclade of *A. subrufescens*, including the type specimen of *A. fiardii*; the remaining 17 formed a well-supported sister clade to *A. subrufescens*, which represents the new species described in our taxonomic treatment. 

### 3.2. ITS Types and Characteristic Polymorphisms

The ITS alignment of 66 sequences of *A. subrufescens* and the newly described species revealed 14 polymorphic positions, which are informative to distinguish between the different ITS types. The characteristic polymorphisms of three types of ITS (A, B and C) within *A. subrufescens* are shown in Table 2. The five samples (LAPAM77, LAPAM78, LAPAM100, LAPAM101 and LAPAM103) from the Dominican Republic have typical ITS type A; and the three Mexican samples have typical ITS type B (IE913) and AB (CA603 and LD201903). Additionally, 17 collections of the newly described species differ from all of the ITS sequences of *A. subrufescens* at five polymorphic positions (166, 196, 210, 494 and 518) and were unique to the newly described taxon (Table 2).

In addition to the 14 polymorphic positions characteristic of different ITS types showed in Table 2, there were 46 other polymorphic positions in the alignment. The variant alleles at these positions were mostly found in a single sample. In rare cases, where they were found in two or at most three samples, they always had the same ITS type. 

In order to have a better representation of the relationships between these different ITS types, a UPGMA tree was constructed (Figure 2). The results showed that the different ITS types were clustered into two groups: the new species bearing the new ITS type M belonged to a distinct lineage, and geographically was restricted to North America; another group clustered samples of *A. subrufescens* with the ITS types A, B and C, or their combinations. The sequences with ITS types A, B or AB were mainly found in the Americas or in Europe, except two samples that were from China, and sequences of the ITS type C were found in Hawaii or Asia. In the tree, three samples (CA918, NTF67 and CA487-c6) were not clustered according to their ITS types (per the assignments in Table 2), due to the presence of variant characters other than the characteristic polymorphic positions.

### 3.3. Taxonomy

***Agaricus*** subg. ***Flavoagaricus*** Wasser, Fl. Fung. RSS Ukrainicae: 138. 1980.

***Agaricus*** sect. ***Arvenses*** (Konrad and Maubl. 1927: 58) Konrad and Maubl., *Encycl. Mycol.* 14: 104. 1948.

***Agaricus macrochlamys*** Medel, Garibay-Orijel, Argüelles-Moyao, G. Mata, Kerrigan, Bessette, Geml, Angelini, L.A. Parra and Linda J. Chen, sp. nov. Figure 3, Figure 4, Figure 5 and Figure 6.

**MycoBank:** MB 844135.

**Etymology:** the name is derived from classical Greek μακρός (macros) meaning large, and χλᾰμῠς (chlamys) meaning dress or mantle, referring to the ample annulus close to the stipe apex.

**Holotype: Mexico, Veracruz**, Xalapa, Parque Ecológico El Haya, Colonia Benito Juárez, 19°31′11″ N 96°56′36″ W, in broadleaf subtropical cloud forest near a bamboo, 3 September 2009, leg. R. Medel, **Medel 1961**, **(XAL)**.

**Macroscopic description (Figure 3 and Figure 5)**: **Pileus** 3.9–13.7 cm diam., truncate-convex when young, then hemispherical to plano-convex, finally plane at maturity, usually with a slightly depressed center (never with prominent umbo); surface dull and dry, very variable in appearance and colors, mostly completely covered with brownish fibrils when young, becoming to fibrillose, slightly reddish or hazelnut-brown or violet-brown squamules over a white, grayish or beige background, denser at the disc, forming an entire brown to dark brown center, less towards the margin; in other cases, also completely white, whitish or very light cream with only the disc colored brown to dark brown; in these collections with pale or whitish pileus, the surface appears smooth or slightly fibrillose, in some collections, somewhat rough, never scaly. Margin always exceeding the lamellae. **Lamellae** free, crowed, up to 1 cm broad, intercalated with numerous lamellulae, white to pinkish when young, finally dark brown at maturity, with finely denticulate whitish edge. **Stipe** 4.6–14.9 × 0.5–1.6 cm, cylindrical, sometimes curved with bulbous to sub-bulbous base (1.5–3.5 cm wide), fistulose, with an annulus in its upper third, above the annulus smooth and below the annulus whitish or slightly rosaceous below, at first pruinose then smooth, with a pure white evanescent flocculosity towards the base, slightly yellowing on handling, sometimes with short white rhizomorphs at the base. **Annulus** attached to the upper part of the stipe, superous, pendent, very broad, fragile, membranous to fibrillose-araneose, double, upper surface white smooth to subtly striate radially, lower surface covered by 0.2–0.4 cm floccose-squamulose white or light brownish square patches (thicker toward the pileus margin, conspicuous in younger specimens). **Context** white, firm, when cut unchanging or discoloring slightly yellowish. Odor of almond or anise. **Spore print** dark brown.

**Microscopic description (Figure 4 and Figure 6)**: **Basidiospores** 4.5–5.5–6.9(–7.5) × (3–)3.3–4–5(–5.3) µm, Q = (1.22–)1.29–1.42–1.67(–1.75), broadly to ellipsoid, smooth, brown, thick-walled (0.4–0.6 µm), without apical pore. **Basidia** mostly four-spored, globose, doliiform, shortly clavate or clavate with slightly truncate apex, hyaline, thin-walled, 9–20 × (4.5–)5.5–9 µm, sterigmata up to 3 µm long. **Cheilocystidia** numerous on the entire edge, hyaline or with internal granular dark brown pigment, thin-walled, catenulate (seldom simple) in short chains of two–four globose or ellipsoid elements measuring 5.5–23(–27) × (4–)7–15 µm. **Pleurocystidia** absent. **Lower surface of the annulus** consisting of two types of hyphae, the internal composed of cylindrical elements 2–11 µm wide, the external composed by inflated elements easily disarticulated in spherical to elliptical (or rarely irregular) elements measuring 4–21 × 4–12 µm. **Pileipellis** a cutis consisting of cylindrical hyphae, hyaline or with yellow granular intracellular pigment, 2–12.5 µm wide, the broader the more constricted at the septa. **Clamp-connections** absent.

**Macrochemical reactions**: Schäffer reaction positive, orange to orange-red; KOH positive, yellow.

**Habit, habitat, occurrence and distribution:** Solitary or gregarious, usually among the leaf litter in (sub)tropical broadleaf cloud forests, only one collection among pine needle litter and another in a grassy place, from lowland to the mountains. Common. Recorded only from the Dominican Republic, Mexico and USA.

**Additional specimens examined: Dominican Republic, Puerto Plata**, Sosúa, Puerto Chiquito, in broadleaf tropical forest, 5 December 2012, leg. C. Angelini, JBSD126480 (duplicate in LAPAM40); **Puerto Plata**, Sosúa, Cemetery, in broadleaf tropical forest, 14 December 2014, leg. C. Angelini, JBSD126482 (duplicate in LAPAM49); **Puerto Plata**, Sosúa, Cemetery, in broadleaf tropical forest, 14 December 2014, leg. C. Angelini, JBSD126483 (duplicate in LAPAM50); **Puerto Plata**, Sosúa, Cemetery, in broadleaf tropical forest, 26 December 2014, leg. C. Angelini, JBSD126484 (duplicate in LAPAM52); **Puerto Plata**, Sosúa, Puerto Chiquito, near the abandoned hotel, in broadleaf tropical forest, 23 November 2016, leg. C. Angelini, JBSD130773 (duplicate in LAPAM79); **Puerto Plata**, Sosúa, Puerto Chiquito, near the abandoned hotel, under *Eucalyptus* sp., 23 December 2016, leg. C. Angelini, JBSD130774 (duplicate in LAPAM178); **Mexico**, **Veracruz**, Xalapa, Jardín Botánico Francisco Javier Clavijero, Instituto de Ecología, in broadleaf subtropical cloud forest, 4 September 2012, leg. A.C. Velázquez-Narváez, Velázquez-Narváez 2357 (XAL); **Veracruz**, Coatepec, Campestre San Rafael, Zoncuantla, in broadleaf subtropical cloud forest, 28 October 2012, leg. R. Medel and F.G. Lorea-Hernández, Medel and Lorea-Hernández 2249; **Veracruz**, Xalapa, Jardín Botánico Francisco Javier Clavijero, Instituto de Ecología, Campus 1, cultivated specimen, 12 September 2019, leg. I. Limón Hernández, CUL-IE4012 (XAL); **Veracruz**, Xalapa, Emiliano Zapata, in leaf litter of *Pinus* sp., 2 February 2019, leg. J. Chen, LD201901(FACBA); **Veracruz**, Xalapa, Jardín Botánico Francisco Javier Clavijero, Instituto de Ecología, Campus 1, in broadleaf subtropical cloud forest, 11 July 2019, leg. J. Chen, LD201927 (XAL); **Veracruz**, Xalapa, Jardín Botánico Francisco Javier Clavijero, Instituto de Ecología, Campus 1, in broadleaf subtropical cloud forest, 12 August 2019, leg. J. Chen, LD201934 (XAL); **Veracruz**, Xalapa, Jardín Botánico Francisco Javier Clavijero, Instituto de Ecología, Campus 1, cultivated specimen, 5 August 2019, leg. I. Limón Hernández, CUL-IE904 (FACBA); **Veracruz**, Xalapa, Jardín Botánico Francisco Javier Clavijero, Instituto de Ecología, Campus 1, cultivated specimen, 6 August 2019, leg. I. Limón Hernández, CUL-IE903 (FACBA); **Veracruz**, Xalapa, Jardín Botánico Francisco Javier Clavijero, Instituto de Ecología, Campus 1, in broadleaf subtropical cloud forest, 12 September 2019, leg. J. Chen, LD201938 (XAL). **USA**, **Georgia**, Camden Co., Sempervirens Trail, Crooked River State Park, 21 April 2015, leg. A.E. Bessette, ARB1353, on broadleaf subtropical coastal forest (private herbarium of Richard W. Kerrigan).

***Agaricus subrufescens*** Peck, Ann. Rep. New York State Mus. 46: 105. 1893. Figure 7, Figure 8 and Figure 9.

MycoBank: MB 248259

= *Agaricus bambusae* Beeli, Bull. Soc. Roy. Bot. Belg. 61(1): 93. 1928.

= *Agaricus*
*fiardii* Pegler, Kew. Bull. Add. Ser. 9: 447. 1983.

= *Agaricus rufotegulis* Nauta, Persoonia 17(2): 231. 1999.

= *Agaricus brasiliensis* Wasser, Didukh, Amazonas and Stamets, Int. J. Med. Mush. 4: 274: 2002. (nom. illeg. non *Agaricus brasiliensis* Fr., Linnaea 5: 509. 1830).

= *Agaricus rufotegulis* var. *hadriaticus* Lancon. and Nauta, Boll. Grupo Micol. G. Bresadola, Nuova Serie 47(2): 17. 2004.

= *Agaricus subrufescens* var. *hadriaticus* (Lancon. and Nauta) E. Ludw., Pilzkompendium 2 (beschreibungen): 62. 2007.

= *Agaricus albopersistens* Zuccher., *I Funghi delle Pinete delle Zone Mediterranee*: 365. 2006.

− *Agaricus blazei* Murrill sensu Heinem. (and auct. plur.), Bull. Jard. Bot. Belg. 62: 365. 1993.

**Etymology:** from the Latin compound *subrufescens* meaning slightly (*sub*-) reddening (*rufescens*), it should presumably refer to the pileus “varying in color from white to grayish to dull reddish-brown”, described by Peck in the original description as etymology is not explained by this author.

**Lectotype: USA, New York,** Glen Cove, L. I., October, coll. W. Falconer, det. C. H. 

Peck, specimen marked “B”. NYS.

**Note:** The following morphological description is based on some unpublished collections of *A. subrufescens* sympatric with *A. macrochlamys* from Mexico and the Dominican Republic. Other sympatric collections of *A. subrufescens* with *A. macrochlamys* were also described previously in Parra et al. [11].

**Macroscopic description (**Figure 7**): Pileus** 5.3–11 cm diam., conical, then hemispherical or trapezoid, finally plano-convex, sometimes depressed at the disc, background whitish covered by fine hazelnut-brown, reddish-brown, brown-violet fibrils becoming squamules toward the disk, forming an entire densely reddish-brown center. Surface dull and dry, slightly yellowing on handling. Margin exceeding the lamellae, even in mature basidiomata. **Lamellae** free, crowded, straight, 0.6–0.8 cm broad, intercalated with numerous lamellulae, at first white (pinkish on handling, resembling a *Leucoagaricus* sp.), then pale pink for a long time, finally dark brown in mature basidiomata, with even and concolors or finely denticulate edge. **Stipe** 9–16 × 0.7–1.4 cm, cylindrical, sometimes curved at the base, enlarged, bulbous or abruptly marginate bulbous at the base (1.6–2.6 cm), fistulose, with an annulus in its upper third; above annulus white and smooth, below annulus white, pruinose when young, frequently covered by a white floccosity particularly towards the base; slightly yellowing on handling. At times with a single thick rhizomorph at the base. **Annulus** superous, double, white, fragile and fine, upper surface smooth to fibrillose-cobweb, lower surface covered by white or ochre floccose to wooly-floccose radially arranged square patches. **Context** in pileus and stipe firm, when cut white, sometimes slightly yellowing, with strong odor of anise. 

**Microscopic description (**Figure 8**): Spores** (4.3–)4.6–5.3–6.2 (–6.9) × 3–3.8–4.5(–4.9) μm, Q = 1.21–1.40–1.67, ellipsoid, brown, thick-walled. **Basidia** tetrasporic, clavate to broadly clavate, hyaline, thin-walled, 13–18 × 5.5–8 μm, sterigmata up to 2 µm long. **Cheilocystidia** 6–18.5(–21) × 6–13(–16) µm, abundant, spherical, ovoid or ellipsoid, catenulate or rarely simple, hyaline, thin-walled. **Pleurocystidia** not observed. **Annulus** consisting of two types of hyphae, the internal composed of cylindrical elements 3.5–15 µm wide, the external composed by spherical or slightly ovoid or inflated elements easily disarticulated, 10–32 × 7–16(–20) µm. **Pileipellis** a cutis consisting of cylindrical hyphae 3–12 µm diam., the wider the more constricted at septa. **Clamp-connections** not observed.

**Macrochemical reactions:** Schäffer reaction positive, orange-red; KOH positive, yellow.

**Habit, habitat, occurrence and distribution:** Gregarious, rarely solitary, usually among the leaf litter in (sub)tropical broadleaf cloud forests. Very common. Cosmopolitan.

**Specimens examined: Dominican Republic**, **Puerto Plata**, Sosúa, Puerto Chiquito, near the abandoned hotel, in broadleaf tropical forest, 27 November 2016, leg. C. Angelini, LAPAM77; **Puerto Plata**, Sosúa, Puerto Chiquito, near the abandoned hotel, in broadleaf tropical forest, 29 November 2016, leg. C. Angelini, LAPAM78; **Puerto Plata**, Sosúa, Puerto Chiquito, near the abandoned hotel, in broadleaf tropical forest, 12 December 2017, leg. C. Angelini, LAPAM100; **Puerto Plata**, Sosúa, Puerto Chiquito, near the abandoned hotel, in broadleaf tropical forest, 12 December 2017, leg. C. Angelini, LAPAM101; **Puerto Plata**, Sosúa, Puerto Chiquito, near the abandoned hotel, in broadleaf tropical forest, 17 December 2017, leg. C. Angelini, LAPAM103. **Mexico**, **Veracruz**, Xalapa, Jardín Botánico Francisco Javier Clavijero, in broadleaf subtropical cloud forest, 31 January 2012, leg. A.C. Velázquez-Narváez, Velázquez-Narváez 2289 (XAL); **Veracruz**, Xalapa, Jardín Botánico Francisco Javier Clavijero, Instituto de Ecología, Campus 1, in broadleaf subtropical cloud forest, 21 March, 2019, leg. J. Chen, LD201903 (FACBA).

**Notes:** From a morphological point of view, it was not possible to find any macroscopic or microscopic differences among *A. macrochlamys*, *A. fiardii* and *A. subrufescens*. Macroscopically, the three species are highly variable in the color of the pileus surface, ranging from white to purplish brown. The Caribbean collections with the white pileus of *A. subrufescens* were described in Parra et al. [11] for the Dominican Republic occurring at the same localities where *A. macrochlamys* was also collected, and were also described in *A. fiardii* by Pegler [18] at varietal rank as *A. fiardii* var. *exalbidus* Pegler, which only differs from the autonymic variety by “the white pileal surface”. Collections of *A. subrufescens* having a white pileus surface have also been published in the United States [14], and the specimen of *A. macrochlamys* from Georgia (USA) included in this study also has a white pileus surface. The annulus is also variable in structure and consistency, varying from fibrillose-araneous and very fragile to membranous and more persistent, but always with the characteristic small floccose to woolly-floccose squamules covering the whole lower surface. Microscopically, in our microscopic examination of the type of *A. fiardii* (F353, K(M) 234316; Figure 9) the following measurements were obtained: Basidiospores 5–5.6–6.1 × 3.6–3.8–4 µm, Q = 1.39–1.47–1.55; basidia 15–25 × 6–9 µm with sterigmata up to 2.5 µm long; and cheilocystidia in short chains of two–four globose or ellipsoid elements 6.5–15 × 6–12 µm. Therefore, all of the available microscopic characters are very similar and overlap widely in the three species, and no relevant distinguishing microscopic characters were noticed among the three species.

For all of the above, the samples from the collections of the three species were sequenced to compare their molecular characters. We observed that the ITS sequence of the type specimen of *A. fiardii* could not be differentiated from *A. subrufescens*, and the former was therefore included under the synonymy of *A. subrufescens* in this work.

However, *A. macrochlamys* shows significant differences with *A. subrufescens* in the ITS (five positions) and *tef1α* regions (six positions and a 4-bps insertion), despite the fact that both species are sympatric in Mexico and the Dominican Republic and the unusual intraspecific variability in *A. subrufescens*. Within *A*. sect. *Arvenses*, *A. macrochlamys* has four species-specific ITS markers: tgtgaGagctt @ 172; atggaGtctc[-] @ 202; ttggaAtgtgg @ 502; ctt[-]tGyggkg @ 528. No specimens having allelic ITS sequences characteristic of both *A. subrufescens* and *A. macrochlamys* are known, strongly implying that they are reproductively isolated, as well as being phylogenetically distinct.

## 4. Discussion

The molecular data from twenty-four samples that were morphologically highly similar were identified as follows: two and five samples from Mexico and the Dominican Republic, respectively, belong to *A. subrufescens*; the remaining samples from Mexico (10), the Dominican Republic (6) and the USA (1) belong to the new species *A. macrochlamys*. In addition, *A. fiardii* is now considered a new synonym of *A. subrufescens*, based on the molecular phylogenetic analyses presented in this paper. 

According to our combined phylogenetic analyses, the new species *A. macrochlamys* formed a full-supported sister clade to *A. subrufescens*. Because no significant morphological differences were observed between the two species, the molecular data remain essential for species identification. Indeed, in comparison with *A. subrufescens*, the new species has five distinguishing ITS nucleotide substitutions, of which four are unique within the *A.* sect. *Arvenses*. This level of genetic divergence is higher than is generally observed between the closely related species in the genus [42]. 

The biology of *Agaricus macrochlamys* remains to be studied in detail. Its geographical range in Mexico and the Dominican Republic seems to be restricted to cloud forests (an ecosystem characterized by temperate and tropical vegetation elements with a high rainfall and subtropical weather). However, the ARB1353 sample from Georgia, USA, was collected in an area with subtropical weather. The life cycles of *A. macrochlamys* need to be compared with those of *A. subrufescens*, which is known to be amphithallic (i.e., both heterothallic and pseudo-homothallic) and versatile [16,43]. Other important questions to be solved are whether this easily cultivated taxon indeed possesses culinary and/or medicinal or nutraceutical value.

## Figures and Tables

**Figure 1 jof-08-00664-f001:**
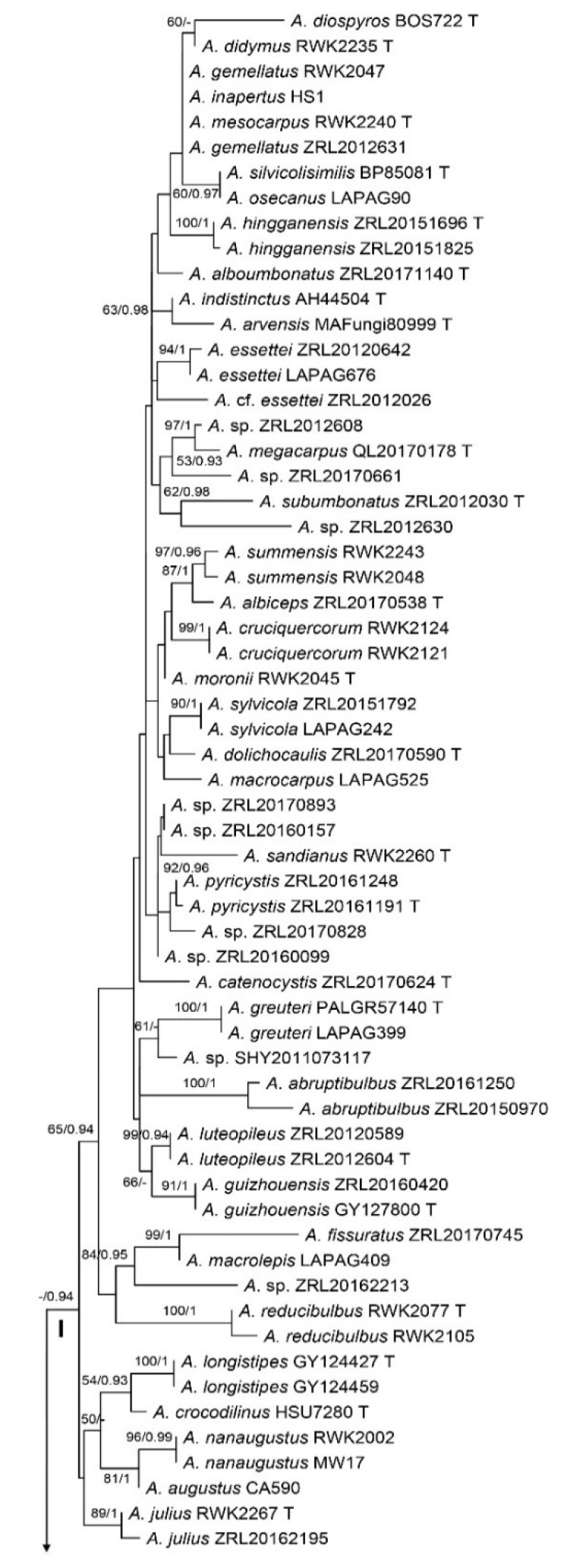
Maximum likelihood (ML) phylogram of *Agaricus* sect. *Arvenses* based on ITS and *tef1* sequence data. Bootstrap support values and Bayesian posterior probabilities greater than 50/0.9 are indicated on branches. New sequences are annotated in red and new species are in bold. T = Type specimen. * CA1110 is the strain isolation of the holotype.

**Figure 2 jof-08-00664-f002:**
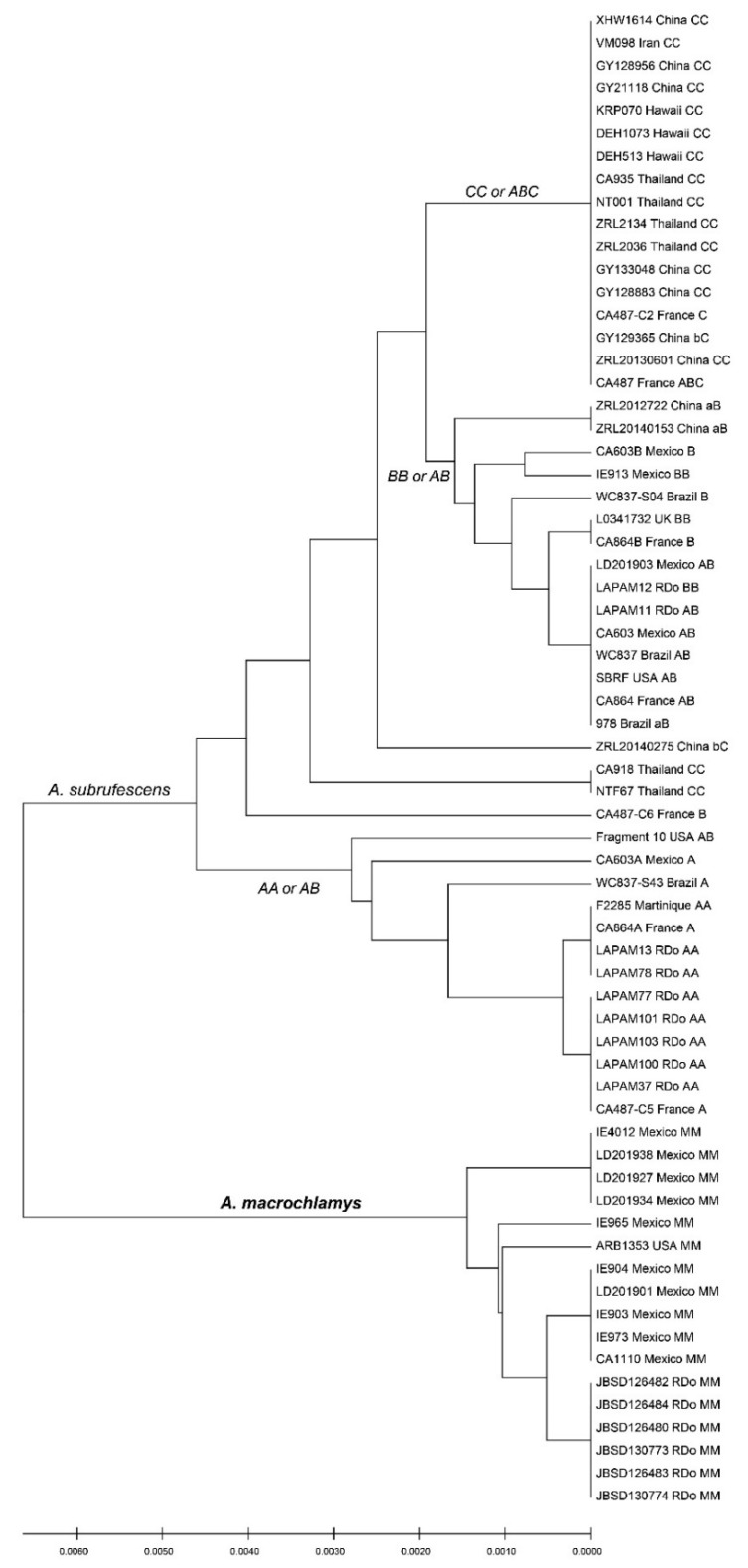
UPGMA analysis of 66 ITS sequences of *A. subrufescens* and *A. macrochlamys*. Lowercase letters in ITS type indicate that the characteristic polymorphisms of the type are not all present.

**Figure 3 jof-08-00664-f003:**
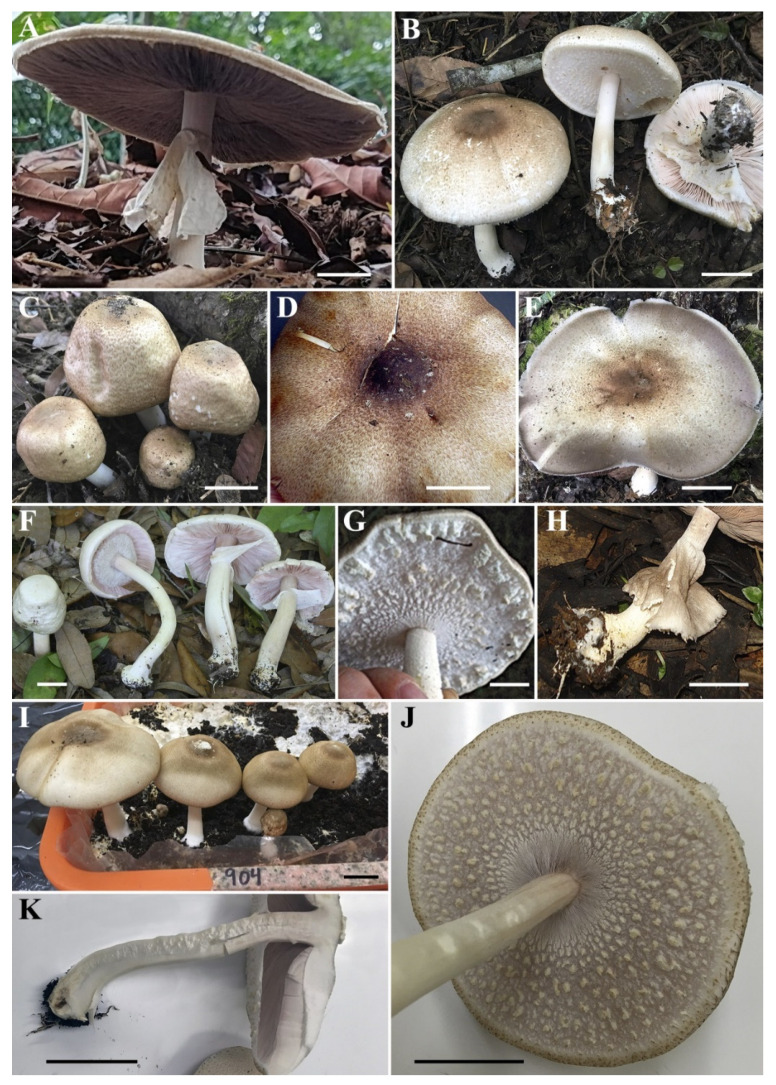
Macroscopic characters of *Agaricus macrochlamys* from Mexico and USA. (**A**–**K**) Basidiomata. (**A**) Medel 1961, holotype; (**B**) LD201938; (**C**) LD201927; (**D**,**G**,**H**) Medel and Lorea-Hernández 2249; (**E**) LD201934; (**F**) ARB1353; (**I**–**K**) CUL-IE904. Bar = 2 cm. Photos by Lorea-Hernández (**A**,**H**); R. Medel (**D**,**G**); J. Chen (**B**,**C**,**E**,**I**–**K**); and A.E. Bessette (**F**).

**Figure 4 jof-08-00664-f004:**
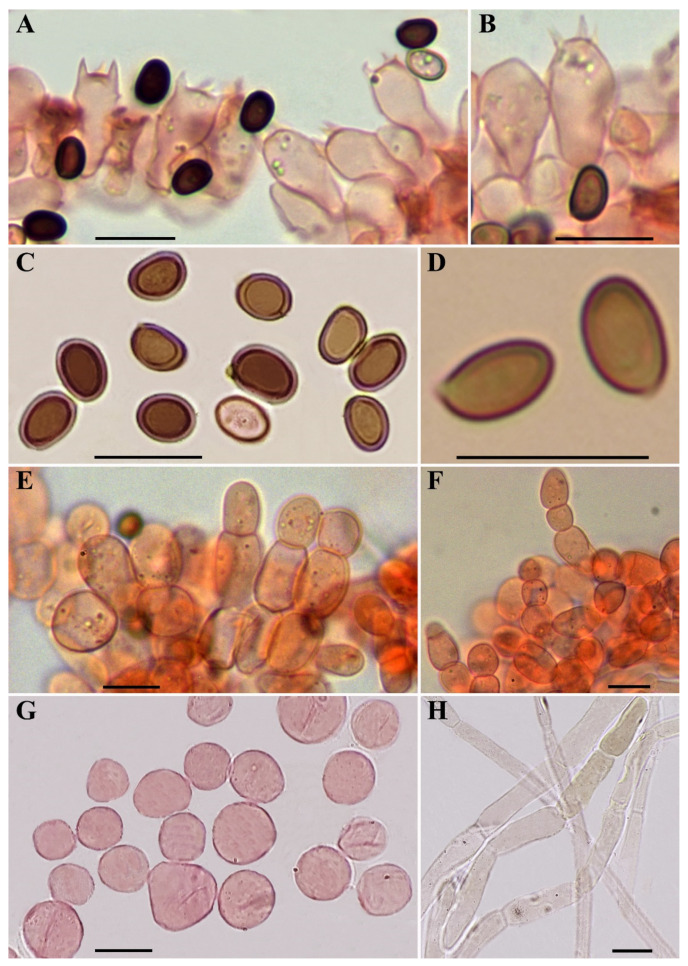
Microscopic characters of *Agaricus macrochlamys* from México and USA. (**A**–**H**) In ammonical Congo red. (**A**,**B**) CUL-IE4012; (**C**,**E**,**F**) CUL-IE903; (**D**) Medel 1961, holotype; (**G**,**H**) LD201938. (**A**,**B**) Basidia; (**C**,**D**) Spores; (**E**,**F**) Cheilocystidia; (**G**) Hyphae of the lower surface of the annulus; (**H**) Pileipellis hyphae. Bar = 10 μm. Photos by J. Chen (**A**–**C**,**E**–**H**) and R. Medel (**D**).

**Figure 5 jof-08-00664-f005:**
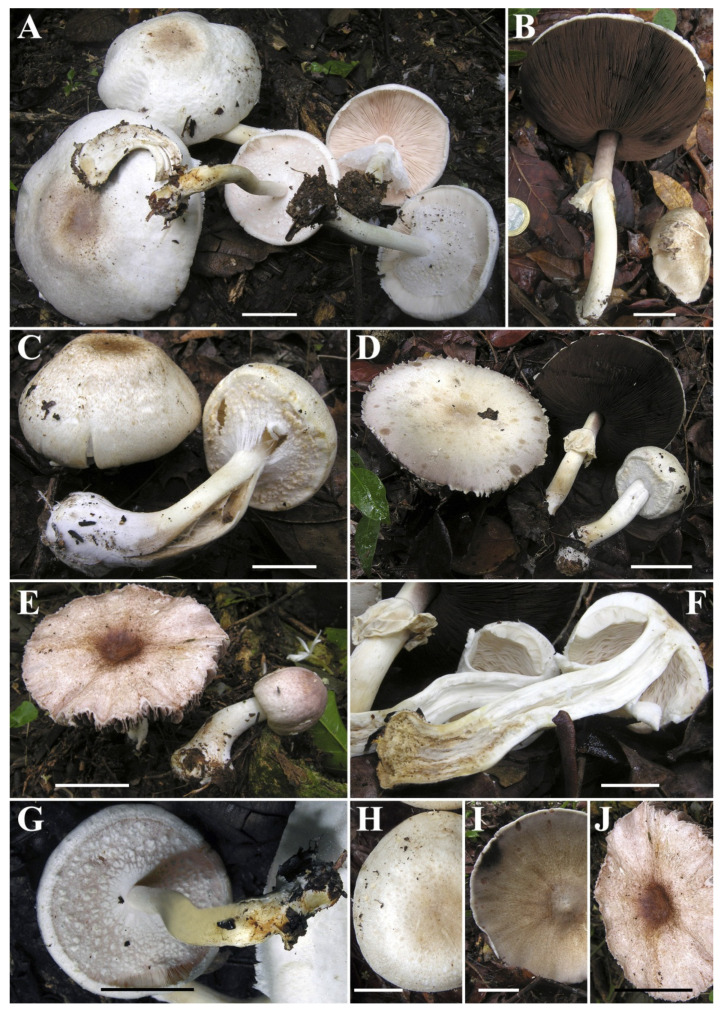
Macroscopic characters of *Agaricus macrochlamys* from the Dominican Republic. (**A**–**J**) Basidiomata. (**A**,**G**) JBSD126480; (**B**,**I**) JBSD126482; (**C**,**H**) JBSD126483; (**D**,**F**) JBSD126484; (**E**,**J**) JBSD130773. Bar = 2 cm. Photos by C. Angelini.

**Figure 6 jof-08-00664-f006:**
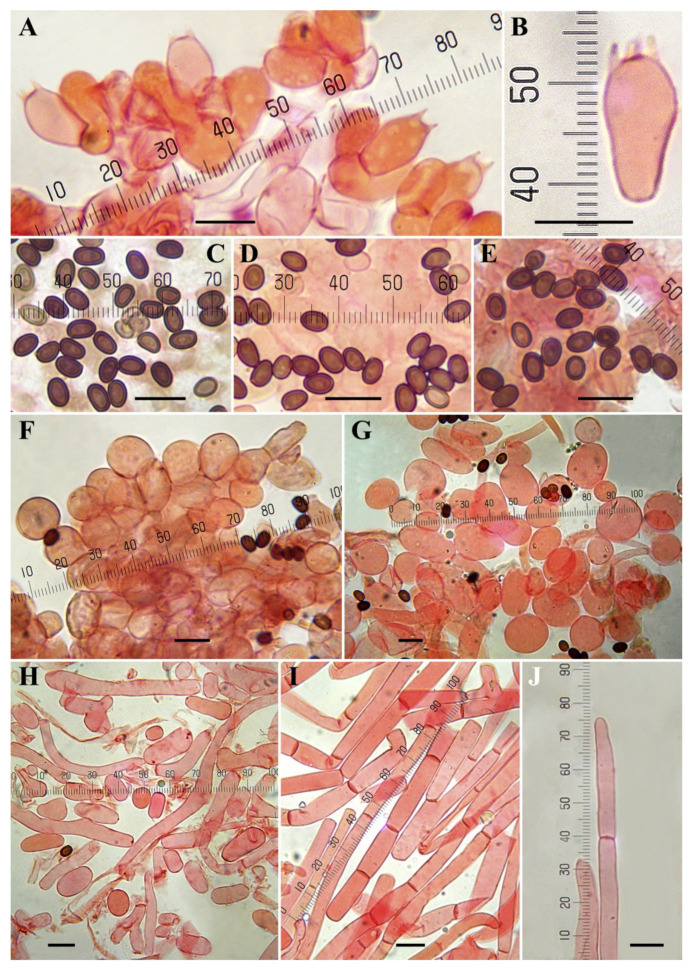
Microscopic characters of *Agaricus macrochlamys* from the Dominican Republic. (**A**–**J**) In ammonical Congo red. (**A**,**F**) JBSD126484; (**B**,**C**) JBSD126480; (**E**,**H**–**J**) JBSD126482; (**D**,**G**) JBSD130773; (**A**,**B**) Basidia; (**C**–**E**) Spores; (**F**,**G**) Cheilocystidia; (**H**) Hyphae of the lower surface of the annulus; (**I**,**J**) Pileipellis hyphae; (**J**) Terminal element. Bar = 10 μm. Photos by L.A. Parra.

**Figure 7 jof-08-00664-f007:**
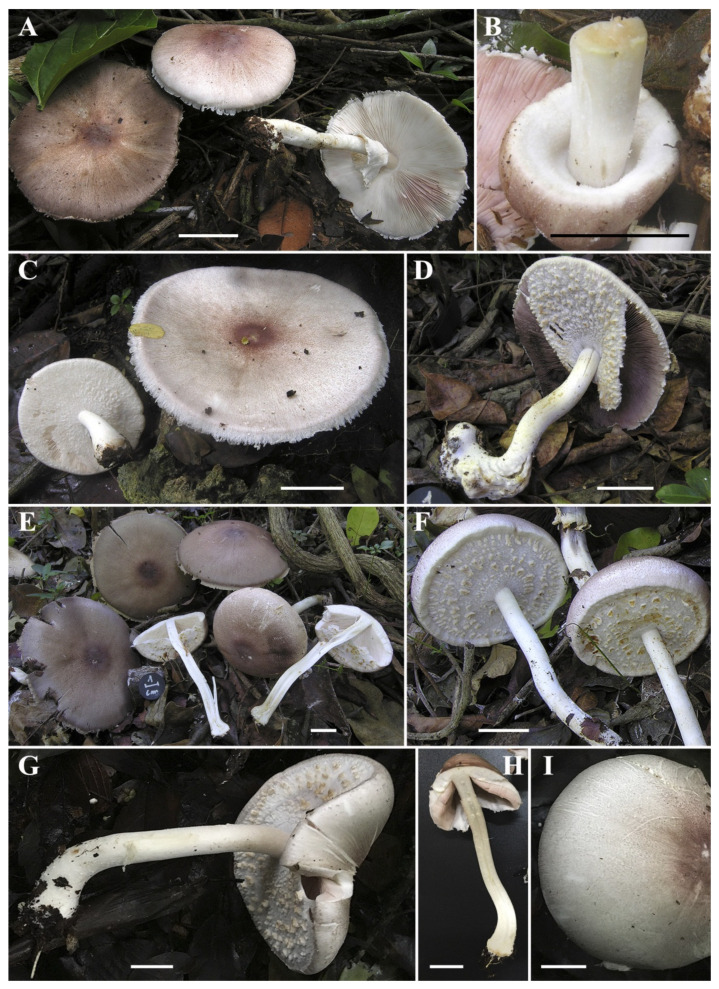
Macroscopic characters of *Agaricus subrufescens*. (**A**–**I**) Basidiomata. (**A**) LAPAM77; (**B**,**C**) LAPAM78; (**D**) LAPAM100; (**E**,**F**) LAPAM103; (**G**–**I**) LD201903. Bar = 2 cm. Photos by C. Angelini (**A**–**F**) and J. Chen (**G**–**I**).

**Figure 8 jof-08-00664-f008:**
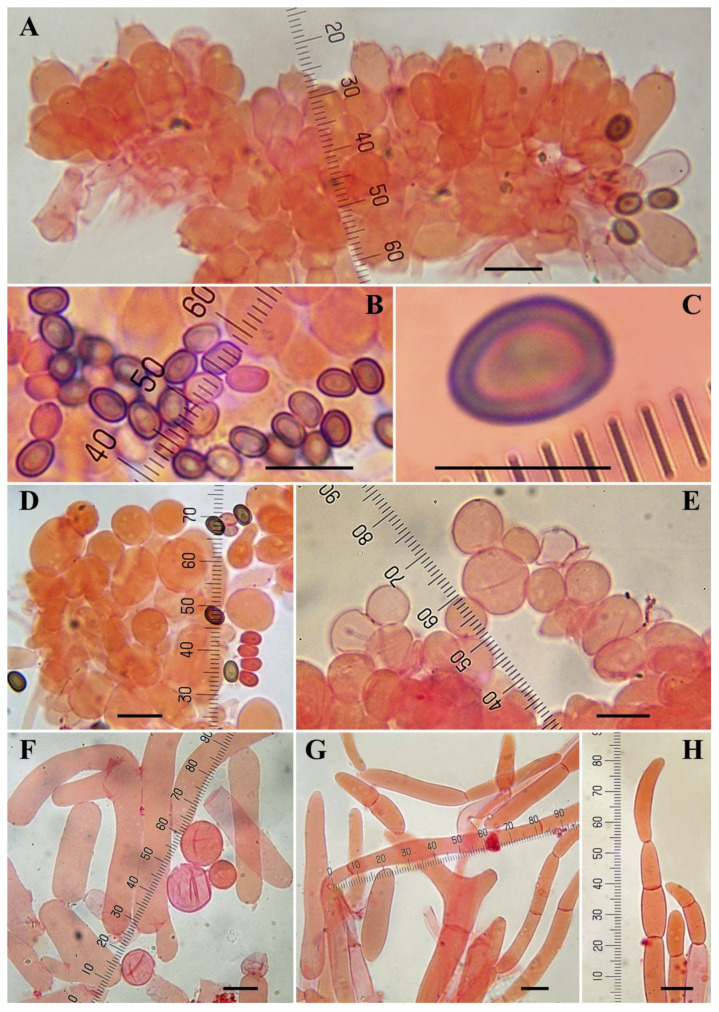
Microscopic characters of *Agaricus subrufescens* from Dominican Republic. (**A**–**H**) In ammonical Congo red (LAPAM78). (**A**) Basidia; (**B**,**C**) Spores; (**D**,**E**) Cheilocystidia; (**F**) Hyphae of the lower surface of the annulus; (**G**,**H**) Pileipellis hyphae; (**H**) Terminal element. Bar = 10 μm. Photos by L.A. Parra.

**Figure 9 jof-08-00664-f009:**
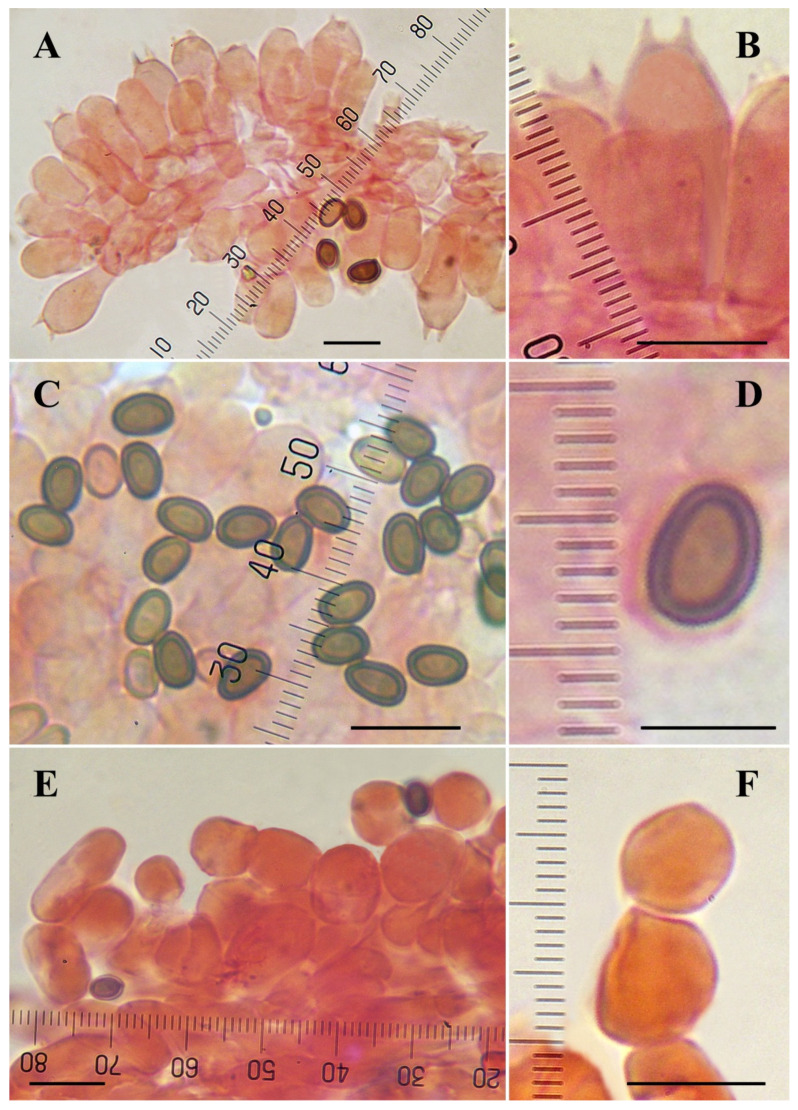
Microscopic characters of type specimen of *Agaricus fiardii*. (**A**–**F**) In ammonical Congo red (F353, K[M] 234316, holotype). (**A**,**B**) Basidia; (**C**,**D**) Spores; (**E**,**F**) Cheilocystidia. (**A**–**C**,**E**,**F**) Bar = 10 μm; (**D**) Bar = 5 μm. Photos by L.A. Parra.

**Table 1 jof-08-00664-t001:** GenBank accession numbers and specimens used in the phylogenetic analyses. New species, new specimens, and new sequences are in bold. T = Type specimen. * CA1110 is the strain isolation of the holotype. ** *A. fiardii*, a new synonym of *A. subrufescens* introduced in this study.

Species	Specimen	ITS	*tef1*	Geographic Origin
*A. abruptibulbus*	ZRL20150970	MK617880	MK614388	Yunnan, China
*A. abruptibulbus*	ZRL20161250	MK617909	MK614417	Jilin, China
*A. albiceps*	ZRL20170538 T	MK617917	MK614424	Inner Mongolia, China
*A. alboumbonatus*	ZRL20171140 T	MK617943	MK614449	Heilongjiang, China
*A. arvensis*	LAPAG450 T	KF114474	KX198047	Spain
*A. augustus*	CA590	JF797193	MK614368	France
*A. catenocystis*	ZRL20170624 T	MK617925	MK614432	Inner Mongolia, China
*A.* cf. *essettei*	ZRL2012026	KT951363	KT951630	Yunnan, China
*A. crocodilinus*	HSU7280 T	NR144989	–	California, USA
*A. cruciquercorum*	RWK2121	KJ859120	–	California, USA
*A. cruciquercorum*	RWK2124	KJ859127	–	California, USA
*A. didymus*	RWK2235 T	NR144990	–	New Mexico, USA
*A. diospyros*	BOS722 T	NR144993	–	Wisconsin, USA
*A. dolichocaulis*	ZRL20170590 T	MK617923	MK614430	Inner Mongolia, China
*A. eburneocanus*	H7528 T	JF495062	–	Australia
*A. essettei*	LAPAG676	KJ548131	–	Savona, Italy
*A. essettei*	ZRL20120642	MK617870	MK614379	Tibet, China
*A. fissuratus*	ZRL20170745	MK617935	MK614441	Inner Mongolia, China
*A. flocculosipes*	ZRL2012105	KT951365	KT951618	Yunnan, China
*A. flocculosipes*	ZRL3028 T	JN664954	–	Doi Inthanon, Thailand
*A. gemellatus*	ZRL2012631	KT951380	KT951623	Tibet, China
*A. gemellatus*	RWK2047 T	KJ859080	–	Utah, USA
*A. greuteri*	LAPAG399	MK617859	–	Palencia, Spain
*A. greuteri*	PAL-GR57140 T	KF114473	–	Emilia-Romagna, Italy
*A. guizhouensis*	ZRL20160420	MK617906	MK614414	Heilongjiang, China
*A. guizhouensis*	GY127800 T	KJ755658	–	Guizhou, China
*A. hingganensis*	ZRL20151825	MK617898	MK614406	Inner Mongolia, China
*A. hingganensis*	ZRL20151696 T	MK617882	MK614390	Heilongjiang, China
*A. inapertus*	HS1	KJ859088	–	California, USA
*A. indistinctus*	AH44504 T	KF114475	–	Burgos, Spain
*A. julius*	ZRL20162195	MK617915	MK614422	Gansu, China
*A. julius*	RWK2267 T	NR144991	–	New Mexico, USA
*A. longistipes*	GY124459	KJ755642	–	Guizhou, China
*A. longistipes*	GY124427 T	KJ755643	–	Guizhou, China
*A. luteopileus*	ZRL20120589	MK617868	MK614377	Tibet, China
*A. luteopileus*	ZRL2012604 T	KT951375	KT951620	Tibet, China
*A. macrocarpus*	LAPAG525	MK617860	MK614369	Bohemia, Czech Republic
** *A. macrochlamys* **	**ARB1353**	**KY704305**	–	Georgia, USA
** *A. macrochlamys* **	**CA1110 ***	**KY704307**	–	Veracruz, Mexico
** *A. macrochlamys* **	**IE903**	**ON332803**	–	Veracruz, Mexico
** *A. macrochlamys* **	**IE904**	**ON332804**	–	Veracruz, Mexico
** *A. macrochlamys* **	**IE965**	**KY114573**	–	Veracruz, Mexico
** *A. macrochlamys* **	**IE973**	**KY114574**	–	Veracruz, Mexico
** *A. macrochlamys* **	**IE4012**	**ON332805**	**ON376243**	Veracruz, Mexico
** *A. macrochlamys* **	**JBSD126480**	**ON332793**	**ON376239**	Puerto Plata, Dominican Republic
** *A. macrochlamys* **	**JBSD126482**	**ON332794**	–	Puerto Plata, Dominican Republic
** *A. macrochlamys* **	**JBSD126483**	**ON332795**	**ON376240**	Puerto Plata, Dominican Republic
** *A. macrochlamys* **	**JBSD126484**	**ON332796**	–	Puerto Plata, Dominican Republic
** *A. macrochlamys* **	**JBSD130773**	**ON332797**	**ON376241**	Puerto Plata, Dominican Republic
** *A. macrochlamys* **	**JBSD130774**	**ON332798**	**ON376242**	Puerto Plata, Dominican Republic
** *A. macrochlamys* **	**LD201901**	**ON332799**	**ON376244**	Veracruz, Mexico
** *A. macrochlamys* **	**LD201927**	**ON332800**	–	Veracruz, Mexico
** *A. macrochlamys* **	**LD201934**	**ON332801**	–	Veracruz, Mexico
** *A. macrochlamys* **	**LD201938**	**ON332802**	**ON376245**	Veracruz, Mexico
*A. macrolepis*	LAPAG409	KJ548128	–	Burgos, Spain
*A. megacarpus*	QL20170178 T	MK617861	MK614371	Gansu, China
*A. megalocarpus*	Hao0608 T	KJ755638	–	Yunnan, China
*A. mesocarpus*	RWK2240 T	KJ859094	–	Colorado, USA
*A. moronii*	RWK2045 T	NR144992	–	Utah, USA
*A. nanaugustus*	MW17	KJ859117	–	Pennsylvania, USA
*A. nanaugustus*	RWK2002	KJ859116	–	Pennsylvania, USA
*A. ornatipes*	MCVE29263 T	MG664795	–	Italy
*A. osecanus*	LAPAG90	KJ548133	–	Madrid, Spain
*A. pyricystis*	ZRL20161248	MK617908	MK614416	Jilin, China
*A. pyricystis*	ZRL20161191 T	MK617907	MK614415	Jilin, China
*A. reducibulbus*	RWK2105	KJ859130	–	Pennsylvania, USA
*A. reducibulbus*	RWK2077 T	NR144994	–	Pennsylvania, USA
*A. sandianus*	RWK2260 T	NR144995	–	New Mexico, USA
*A. silvicolisimilis*	BP85081 T	KF114472	–	Budapest, Hungary
*A.* sp.	AMZ	AY484687	–	Hungary
*A.* sp.	SHY2011073117	KT951407	KT951622	Yunnan, China
*A.* sp.	Thoen6951	JN204433	–	Senegal
*A.* sp.	Thoen7297	JF514542	–	Senegal
*A.* sp.	TL1644	JF495060	–	Australia
*A.* sp.	TL1650	JF495061	–	Australia
*A.* sp.	ZRL2012608	KT951377	KT951627	Tibet, China
*A.* sp.	ZRL2012630	KT951379	KT951621	Tibet, China
*A.* sp.	ZRL20160099	MK617903	MK614411	Heilongjiang, China
*A.* sp.	ZRL20160157	MK617904	MK614412	Heilongjiang, China
*A.* sp.	ZRL20162213	MK617916	MK614423	Beijing, China
*A.* sp.	ZRL20170661	MK617927	MK614434	Inner Mongolia, China
*A.* sp.	ZRL20170828	MK617941	MK614447	Heilongjiang, China
*A.* sp.	ZRL20170893	MK617942	MK614448	Heilongjiang, China
*A. subantarcticus*	GAL9419 T	DQ232642	–	New Zealand
*A. subrufescens*	ZRL2012722	KT951383	KT951632	Yunnan, China
*A. subrufescens*	ZRL20140153	MK617874	MK614382	Yunnan, China
*A. subrufescens ***	K(W)234316	**ON561779**	–	Martinique
*A. subrufescens*	**LD201903**	**ON332806**	**ON376236**	Veracruz, Mexico
*A. subrufescens*	**LAPAM11**	MF511113	**ON376237**	Puerto Plata, Dominican Republic
*A. subrufescens*	**LAPAM12**	MF511114	**ON376238**	Puerto Plata, Dominican Republic
*A. subrufescens*	LAPAM13	MF511115	–	Puerto Plata, Dominican Republic
*A. subrufescens*	LAPAM37	MF511128	–	Puerto Plata, Dominican Republic
*A. subrufescens*	**LAPAM77**	**ON332807**	**ON568203**	Puerto Plata, Dominican Republic
*A. subrufescens*	**LAPAM78**	**ON332808**	–	Puerto Plata, Dominican Republic
*A. subrufescens*	**LAPAM100**	**ON332809**	–	Puerto Plata, Dominican Republic
*A. subrufescens*	**LAPAM101**	**ON332810**	–	Puerto Plata, Dominican Republic
*A. subrufescens*	**LAPAM103**	**ON332811**	–	Puerto Plata, Dominican Republic
*A. subtilipes*	MFLU120992	KP705079	–	Thailand
*A. subtilipes*	LD201201 T	KP705078	MK614370	Thailand
*A. subumbonatus*	ZRL2012030 T	KT951364	KT951628	Yunnan, China
*A. summensis*	RWK2048	KJ859105	–	Utah, USA
*A. summensis*	RWK2243	KJ859104	–	Colorado, USA
*A. sylvicola*	LAPAG242	KJ548130	–	Burgos, Spain
*A. sylvicola*	ZRL20151792	MK617894	MK614402	Inner Mongolia, China
*A. edmondoi* (outgroup)	LAPAG412	KT951326	KT951590	Segovia, Spain

**Table 2 jof-08-00664-t002:** Different types of ITS within *Agaricus subrufescens* and *A. macrochlamys* based on characteristic polymorphisms at 14 positions.

Sample	Country	ITS						Position									GenBank
		Type	39	122	130	145	146	166	196	200	210	269	466	475	494	518	
IE965	Mexico	MM	-	G	A	A	T	**G**	**G**	C	**G**	G	A	T	**A**	**G**	KY114573
IE973	Mexico	MM	-	G	A	A	T	**G**	**G**	C	**G**	G	A	T	**A**	**G**	KY114574
CA1110	Mexico	MM	-	G	A	A	T	**G**	**G**	C	**G**	G	A	T	**A**	**G**	KY704307
IE903	Mexico	MM	-	G	A	A	T	**G**	**G**	C	**G**	G	A	T	**A**	**G**	ON332803
IE904	Mexico	MM	-	G	A	A	T	**G**	**G**	C	**G**	G	A	T	**A**	**G**	ON332804
IE4012	Mexico	MM	-	G	A	A	T	**G**	**G**	C	**G**	G	A	T	**A**	**G**	ON332805
LD201901	Mexico	MM	-	G	A	A	T	**G**	**G**	C	**G**	G	A	T	**A**	**G**	ON332799
LD201927	Mexico	MM	-	G	A	A	T	**G**	**G**	C	**G**	G	A	T	**A**	**G**	ON332800
LD201934	Mexico	MM	-	G	A	A	T	**G**	**G**	C	**G**	G	A	T	**A**	**G**	ON332801
LD201938	Mexico	MM	-	G	A	A	T	**G**	**G**	C	**G**	G	A	T	**A**	**G**	ON332802
JBSD126480	Dominican	MM	-	G	A	A	T	**G**	**G**	C	**G**	G	A	T	**A**	**G**	ON332793
JBSD126482	Dominican	MM	-	G	A	A	T	**G**	**G**	C	**G**	G	A	T	**A**	**G**	ON332794
JBSD126483	Dominican	MM	-	G	A	A	T	**G**	**G**	C	**G**	G	A	T	**A**	**G**	ON332795
JBSD126484	Dominican	MM	-	G	A	A	T	**G**	**G**	C	**G**	G	A	T	**A**	**G**	ON332796
JBSD130773	Dominican	MM	-	G	A	A	T	**G**	**G**	C	**G**	G	A	T	**A**	**G**	ON332797
JBSD130774	Dominican	MM	-	G	A	A	T	**G**	**G**	C	**G**	G	A	T	**A**	**G**	ON332798
ARB1353	USA	MM	-	G	A	A	T	**G**	**G**	C	**G**	G	A	T	**A**	**G**	KY704305
F2285	Martinique	AA	**T**	**A**	**A**	**G**	**A**	A	A	**T**	A	G	A	T	Y	T	JF797201
LAPAM13	Dominican	AA	**T**	**A**	**A**	**G**	**A**	A	A	**T**	A	G	A	T	T	T	MF511115
LAPAM37	Dominican	AA	**T**	**A**	**A**	**G**	**A**	A	A	**T**	A	G	A	T	C	T	MF511128
LAPAM77	Dominican	AA	**T**	**A**	**A**	**G**	**A**	A	A	**T**	A	G	A	T	C	T	ON332807
LAPAM78	Dominican	AA	**T**	**A**	**A**	**G**	**A**	A	A	**T**	A	G	A	T	Y	T	ON332808
LAPAM100	Dominican	AA	**T**	**A**	**A**	**G**	**A**	A	A	**T**	A	G	A	T	Y	T	ON332809
LAPAM101	Dominican	AA	**T**	**A**	**A**	**G**	**A**	A	A	**T**	A	G	A	T	C	T	ON332810
LAPAM103	Dominican	AA	**T**	**A**	**A**	**G**	**A**	A	A	**T**	A	G	A	T	C	T	ON332811
CA603A	Mexico	A	**T**	**A**	**A**	**G**	**A**	A	A	**T**	A	G	A	T	C	T	KY704306 ^1^
WC837-S43	Brazil	A	**T**	**A**	**A**	**G**	**A**	A	A	**T**	A	G	A	T	C	T	KJ541797
CA864A	France	A	**T**	**A**	**A**	**G**	**A**	A	A	**T**	A	G	A	T	T	T	KU557349 ^1^
CA487-C5	France	A	**T**	**A**	**A**	**G**	**A**	A	A	**T**	A	G	A	T	C	T	KU557352
CA603B	Mexico	B	-	G	G	A	T	A	A	C	A	G	A	T	C	T	KY704306 ^1^
L0341732	UK	BB	-	G	G	A	T	A	A	C	A	G	A	T	C	T	AY818649
IE913	Mexico	BB	-	G	G	A	T	A	A	C	A	G	A	T	C	T	KY114572
LAPAM12	Dominican	BB	-	G	G	A	T	A	A	C	A	G	A	T	C	T	MF511114
WC837-S04	Brazil	B	-	G	G	A	T	A	A	C	A	G	A	T	C	T	KJ541796
CA864B	France	B	-	G	G	A	T	A	A	C	A	G	A	T	C	T	KU557349 ^1^
CA487-C6	France	B	-	G	G	A	T	A	A	C	A	G	A	T	C	T	KU557353
CA487-C2	France	C	-	G	G	A	T	A	A	C	A	**A**	**C**	**-**	C	T	KU557351
GY128883	China	CC	-	G	G	A	T	A	A	C	A	**A**	**C**	**-**	C	T	KJ755634
GY133048	China	CC	-	G	G	A	T	A	A	C	A	**A**	**C**	**-**	C	T	KJ755635
GY121118	China	CC	-	G	G	A	T	A	A	C	A	**A**	**C**	**-**	C	T	KJ755633
GY128956	China	CC	-	G	G	A	T	A	A	C	A	**A**	**C**	**-**	C	T	KJ755632
XHW1614	China	CC	-	G	G	A	T	A	A	C	A	**A**	**C**	**-**	C	T	KJ755636
ZRL2036	Thailand	CC	-	G	G	A	T	A	A	C	A	**A**	**C**	**-**	C	T	KU557345
ZRL2134	Thailand	CC	-	G	G	A	T	A	A	C	A	**A**	**C**	**-**	C	T	KU557346
ZRL20130601	China	CC	-	G	G	A	T	A	A	C	A	**A**	**C**	**-**	C	T	MK617873
VM098	Iran	CC	-	G	G	A	T	A	A	C	A	**A**	**C**	**-**	C	T	KT983412
NT001	Thailand	CC	-	G	G	A	T	A	A	C	A	**A**	**C**	**-**	C	T	KU557347
CA935	Thailand	CC	-	G	G	A	T	A	A	C	A	**A**	**C**	**-**	C	T	KU557348
CA918	Thailand	CC	-	G	G	A	T	A	A	C	A	**A**	**C**	**-**	C	T	KJ541798
NTF67	Thailand	CC	-	G	G	A	T	A	A	C	A	**A**	**C**	**-**	C	T	JF514529
DEH513	Hawaii	CC	-	G	G	A	T	A	A	C	A	**A**	**C**	**-**	C	T	AY818646
DEH1073	Hawaii	CC	-	G	G	A	T	A	A	C	A	**A**	**C**	**-**	C	T	AY818647
KRP070	Hawaii	CC	-	G	G	A	T	A	A	C	A	**A**	**C**	**-**	C	T	AY818648
CA487	France	ABC	T/-	R	R	R	W	A	A	Y	A	R	M	T/-	C	T	KU557351 ^2^ KU557352 ^2^ KU557353 ^2^
CA603	Mexico	AB	T/-	R	R	R	W	A	A	Y	A	G	A	T	C	T	KY704306
LD201903	Mexico	AB	T	R	R	A	T	A	A	Y	A	G	A	T	C	T	ON332806
LAPAM11	Dominican	AB	T	R	R	R	W	A	A	Y	A	G	A	T	Y	T	MF511113
WC837	Brazil	AB	T/-	R	R	R	W	A	A	Y	A	G	A	T	C	T	KU557350
CA864	France	AB	T/-	R	R	R	W	A	A	Y	A	G	A	T	Y	T	KU557349
Fragment 10	USA	(AB)	T/-	R	R	R	W	A	A	Y	A	G	A	T	C	T	EU071699
SBRF	USA	(AB)	T/-	R	R	R	W	A	A	Y	A	G	A	T	C	T	AY818656
ZRL2012722	China	(aB)	-	A	G	A	T	A	A	C	A	G	A	-	C	T	KT951383
ZRL20140153	China	(aB)	-	A	G	A	T	A	A	C	A	G	A	-	C	T	MK617874
978	Brazil	(aB)	-	G	G	R	T	A	A	C	A	G	A	T	C	T	AY818650
GY129365	China	(bC)	-	G	G	A	T	A	A	C	A	R	M	-	C	T	KJ755631
ZRL20140275	China	(bC)	-	G	G	A	T	A	A	C	A	G	A	-	C	T	MK617876

Lowercase letter in an ITS type indicates that the characteristic polymorphisms of the type are not all present. The parenthesis indicates that the haploid ITS sequences of these heterokaryons are unknown. ^1^ Haploid sequences, not deposited individually in GenBank, obtained by interpreting the two sequences switched in the heteromorphic electropherograms; ^2^ Haploid sequences of each type obtained by sequencing the clones of the heterokaryotic strain, only the haploid sequences are deposited in GenBank. Bold letters denote unique nucleotide substitutions for each ITS type. ‘Type’ as recorded here specifically refers to the genotype of the culture, which in heterokaryons will typically be comprised of two allelic copies (or rarely three) which may be homoallelic or heteroallelic, with each allele having its own molecular ‘type’ designated A, B, C or M.

## Data Availability

Not applicable.

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
