# Peer review of "Agaricus macrochlamys, a New Species from the (Sub)tropical Cloud Forests of North America and the Caribbean, and Agaricus fiardii, a New Synonym of Agaricus subrufescens"

_jof, 2022, doi:10.3390/jof8070664_

Round 1

Reviewer 1 Report

This study reported two new Agaricus species from America. The authors used morphological and molecular methods to describe the two new species. I advise the manuscript need adding new necessary analyses.

Comments were listed below:
1. Adding a section for estimation of evolutionary divergence among these Agaricus species sequences. Divergence time can proved the taxonomic status of the two species.
2. The pictures of macroscopic characters should be simplified and all of them need a bar. The pictures of microscopic characters also need adding a bar.

Author Response

1. Thank you for the suggestion. However, we think that estimation of evolutionary divergence time is not necessary in our case. In the genus Agaricus, the divergence time has been proposed as a standard criterion for higher ranks such as subgenera and sections (Zhao et al. 2016). And in our case, we are comparing two closely related species. In all our phylogenetic trees (both single gene, and combined gene tree), they are separated into two sister clades with strong support. The observed nucleotides differences are stable and no specimens having allelic ITS/tef sequences characteristic of both A. subrufescens and A. macrochlamys are known, strongly implying that they are reproductively isolated as well as being phylogenetically distinct. In addition, the new species has five distinguishing ITS nucleotide substitutions, of which four are unique within the A. sect. Arvenses. This level of genetic divergence is higher than what generally is observed between closely related species in the genus Agaricus.

2. Scale bars have been added to all photo plates. We prefer to keep the pictures of macroscopic characters, with the aim of showing important morphological characters and morphological variations at different growth stages and specimens from the same or different countries.

Reviewer 2 Report

There are a little bit errors in the paper.

Author Response

  1. L90, “in” should be added before Mexico.

Done

  1. L100, “respectively” should be added after Montiel 20.

Done

  1. L284, more or less "dark" brown? more or less? How to distinguish "less dark brown" from "more brown"?

“with only the disc colored more or less brown” has been replaced by “with only the disc colored brown to dark brown”

  1. L288, “when young” should be added.

Done

  1. L290, (1.5–3.5 cm) in diam.?

We have corrected to “(1.5–3.5 cm wide)”.

  1. L320 better remove the word “scattered”.

Done

  1. L339, better delete “other material”.

We have replaced “other material examined” by “additional specimens examined”

  1. L388, the reviewer suggested to remove the Etymology from the description of subrufescens.

Thanks for the suggestion, however, we prefer to keep it for two reasons: 1) for the consistency; 2) the etymology of this species has never been explained before.

  1. L402, a comma is need after “dull and dry”.

Done

  1. L406, 0.70 should be written as 0.7

Done

  1. L412, smooth and fibrillose?

We have corrected to “smooth to fibrillose-cobweb”.

  1. L430, detele “material” in material examined.

We have corrected to “Specimens examined”

  1. There are a few typing errors (some words were separated by a hyphen) due to the conversion of the file.

Many thanks for the observations. However, we have double checked our word document, and we haven´t seen the same problem. Also, this was not observed by another reviewer.  

Reviewer 3 Report

Dear authors and editors,

Here is  the review  of the paper titled "Agaricus macrochlamys, a new species from the (sub)tropical cloud forests of North America and the Caribbean, and Agaricus fiardii, a new synonym of Agaricus subrufescens" written by Rosario Medel-Ortiz and her co-authors.

The paper aims to give taxonomic description of Agaricus macrochlamys, a new species to science.It is a species morphologically identical to a widespread A. subrufescens. Many collections of both species were analyzed based on their morphology and molecular characters including the holotype of A. fiardii, another enigmatic taxon from this group. The authors did not find any reliable distinguishing morphological characters that could be use for  species delimitation between A. subrufescens and the newly described A. macrochlamys. On the other hand, both species can be mutaually distinguished by by several ITS and tef1α species-specific markers and a 4-bp insertion in the tef1α sequence.The study confirmed A. fiardii as a new synonym of A. subrufescens.

The paper is mostly well done! Morphological description, phylogenetic study and discussion are exhaustive and cover all needed parts. The English language is very good. The authors followed the newest version of International code of nomenclature for algae, fungi, and plants. The major suggestion is to include original references (in Table 1.) for all sequences used in this study (see pdf). In that way you'll give a credit to the scientists who sequenced samples used for molecular analysis in this study.

Also, please go through a few additional minor remarks included in the revised pdf version of the manuscript (attached) that would give additional quality to the paper.

Best, reviewer

Author Response

  1. Please include reference column in the table and for each specimen used in the phylogenetic analysis please add the original reference. It is because it is important to give credit to original authors of the sequences.

Many thanks for the suggestion. The sequences used in our study have been cited in M&M as follows “and the remaining were retrieved from GenBank and were used in previous studies [2,5,31]. The sample’s origin and GenBank accession numbers are listed in Table 1.”

We have checked some recent publications of JOF, to cite the original reference of the sequence is not required. Thus, we prefer to not add one more column to the table, which is already very full.

  1. Please mark sequence obtained from the holotype.

We have added an asterisk to CA1110 and added “* CA1110 is the strain isolation of the holotype.” in the caption of Table 1 at L158.

  1. L275, please include the geographic coordinates of the holotype locality. Also, please include accession no. of the sequence obtained from the holotype here.

The geographic coordinates have been added. However, we could not include the accession no. of the sequence obtained from the holotype because CA1110 is a culture isolate from the holotype. This information has been provided in both captions of Table 1 and Figure 1.

  1. L507 “needs to be” should be corrected to “need to be”.

Done

Round 2

Reviewer 1 Report

I am worried that the difference of only 5 bases is not enough to indicate that the species is new, and it may be a variant. The results of differentiation time should be able to be the key evidence for new species.

Reviewer 3 Report

Dear authors and Editor,

The manuscript is improved according to most of my suggestions and is appropriate for publication in JoF now.

Best, reviewer